# Pilot Testing of a Nudge-Based Digital Intervention (Welbot) to Improve Sedentary Behaviour and Wellbeing in the Workplace

**DOI:** 10.3390/ijerph17165763

**Published:** 2020-08-10

**Authors:** Caitlin Haile, Alison Kirk, Nicola Cogan, Xanne Janssen, Ann-Marie Gibson, Bradley MacDonald

**Affiliations:** Psychological Sciences and Health, Graham Hills Building, University of Strathclyde, Glasgow G1 1QX, Scotland, UK; caitlinhaile@hotmail.co.uk (C.H.); alison.kirk@strath.ac.uk (A.K.); xanne.janssen@strath.ac.uk (X.J.); annmarie.gibson2015@gmail.com (A.-M.G.); bradley.macdonald@strath.ac.uk (B.M.)

**Keywords:** sedentary behaviour, wellbeing, mental health, physical activity, digital health, workplace, behaviour change, occupational health, nudge theory

## Abstract

Welbot is a nudge-based digital intervention that aims to reduce sedentary behaviour and improve physical and mental wellbeing at work. The purpose of this study was to pilot test the Welbot intervention. Forty-one (6M/35F) University staff (*M* age = 43-years) participated in this study, which was a single arm repeated measures trial conducted over three weeks of intervention. The primary outcome was sedentary behaviour (measured subjectively and objectively) and secondary outcomes included: mental wellbeing, procrastination, depression, anxiety and stress, and work engagement. A subset of participants (*n* = 6) wore an ActivPAL to objectively measure activity data, while another subset of participants (*n* = 6) completed a qualitative semi-structured interview to ascertain experiences of using Welbot. Following the intervention, a Friedman non-parametric test revealed that participants self-reported significantly less time sitting and more time standing and objectively recorded more steps at the week-1 follow-up. A series of paired *t*-tests exhibited that changes in all secondary outcomes were in the expected direction. However, only improvements in depression, anxiety, and stress were significant. After using Welbot, thematic analysis demonstrated that participants perceived they had a positive behaviour change, increased awareness of unhealthy behaviours at work, and provided suggestions for intervention improvement. Overall, findings provided indications of the potential positive impact Welbot may have on employees’ wellbeing, however, limitations are noted. Recommendations for intervention improvement including personalisation (e.g., individual preferences for nudges and the option to sync Welbot with online calendars) and further research into how users engage with Welbot are provided.

## 1. Introduction

Research exploring the impact of employee wellbeing in the workplace has advanced substantially over the last two decades. There is growing evidence linking workplace performance and productivity with mental and physical health issues among employees [1]. Research has shown that implementing interventions in the workplace designed to improve employee health and wellbeing have frequently resulted in health status improvements and enhanced work performance [2,3]. While such interventions vary in duration, composition, and intensity, they are all designed to promote an increase in healthy lifestyle behaviours, including stress management, improved nutrition, and reducing sedentary behaviour (defined as any waking behaviour that occurs in a sitting or lying position and results in an energy expenditure of 1.5 METs or lower) [4,5].

As society has progressed and technology has advanced, there has been a decline in demand for manual labour-based jobs, with these being replaced by more office or sedentary-based occupations. In a sample of UK full-time office workers, 65% of time at work was sedentary and sitting at work accounted for 63% of total daily sitting time [6]. With such a large proportion of an adult’s day being spent at work, the importance of reducing sedentary behaviour in the workplace is highlighted. Indeed, sedentary behaviour has a negative impact on health, being associated with an increased risk of diabetes, obesity, cardiovascular problems, as well as mental health issues and some types of cancer [7]. Research suggests metabolic health is compromised in those who spend the majority of their days engaged in sedentary behaviour, even if they are engaging in moderate/vigorous physical activity [8]. In other words, sedentary behaviour is an independent risk factor for obesity and chronic disease. Therefore, interventions designed to target sedentary behaviour in the workplace are specifically needed.

Research suggests that breaking up prolonged sedentary time can have positive health outcomes for an individual. Short-term laboratory-based experiments have reported that when sitting is interrupted every 30 min by brief activity breaks (i.e., two minutes of treadmill walking or light resistance activity), postprandial glucose and insulin levels are significantly reduced [9,10]. Moreover, research has demonstrated a significant relationship between health outcomes (i.e., triglycerides, glucose, waist circumference) and the total number of breaks from sitting, independent of total sedentary time [11]. Research has also demonstrated that implementing interventions designed to reduce sedentary behaviour in the workplace can not only reduce sitting time, but can also indirectly lead to improvements in mental wellbeing (fatigue, tension, depression, and anxiety) [12]. Therefore, when examining interventions, it is important to consider a range of factors that may also be impacted as indirect effects are common.

Recent reviews of interventions to reduce sedentary behaviour in the workplace include strategies such as educational/behavioural, environmental, and multi-component, including the use of computer-based or mobile health technologies [13,14,15]. All of these strategies have shown some success at improving sedentary behaviour and physical and mental wellbeing. Indeed, research has documented that computer-based, mobile, and wearable technology-based interventions targeting workplace sitting, compared to non-work-based sitting, were more effective at medium-term (3–6 months) follow-up [14]. A number of limitations have been documented in these published reviews including limited description of the intervention, in addition to a lack of qualitative measures to explore participant experiences. Cost of intervention delivery is also a limiting factor for large scale implementation and impact.

Digital-based interventions have the potential to reach large populations at a low cost and offer potential to tailor interventions to the needs of individuals or specific groups of office workers. A recent systematic review of occupational-based digital health interventions displayed moderate evidence for interventions in improving employee health (i.e., psychological wellbeing, sleep, physical activity, and sedentary behaviour) [16]. Nevertheless, a large decline in technology use and engagement is often observed with digital interventions [15]. Yet, there is evidence that a nudge or prompt driven approach can yield high engagement in intervention components as well as reduce sitting time and increase light activity at work [17,18].

The aim of this study was to pilot test a nudge-based digital intervention (Welbot) to reduce sedentary behaviour and improve physical and mental wellbeing in the workplace. Unlike many other digital interventions, Welbot has been developed with academic and professional input and in collaboration with industry specialists. It is also a more holistic approach as it aims to target many of the different areas that contribute to one’s overall wellbeing at work (e.g., reducing sedentary behaviour, increasing physical activity, having regular screen breaks, improving mental wellbeing, promoting positive work environments, ensuring that you are staying hydrated, and eating healthily, etc.).

## 2. Materials and Methods

### 2.1. Intervention: Welbot

Welbot is a cross-platform, personalised, digital intervention that aims to reduce sedentary behaviour and improve physical and mental wellbeing at work [19]. It incorporates activities such as stretching, screen breaks, exercises, mindfulness, hydration tips, breathing exercises, etc., which are delivered to users in the form of “nudges” (see Figure 1). A nudge is a notification that asks users to engage in a simple 1–3-min activity (e.g., an arm stretch, a mindful cup of tea, or a screen break) that progresses from a preparation card (i.e., what the nudge will entail), to a doing card (i.e., how to perform the nudge), to a done card (i.e., why the nudge is good for our physical and/or mental wellbeing). These nudges are delivered at regular intervals (every 40 min; or 12 nudges per workday) and are designed to boost an employee’s overall wellbeing. Welbot is a personalised digital intervention as it asks users what goals they would like to prioritise (e.g., “Stand Up, Sit Less, and Move More” or “Reduce Stress”) and modifies content in line with this. It also adapts content according to a number of person-specific characteristics, including: working hours, work environment, and preferences of nudges (i.e., what content the user likes/does not like to engage with). Hence, it takes user’s individual circumstances and preferences into consideration when tailoring its content.

A UK University has been working closely with the Welbot company to further develop the digital intervention. The ultimate aim of this collaboration was to create a digital intervention that was evidence-based and uniquely tailored to each individual user. Through this collaboration, the University research team had 3 primary roles: (1) to validate and evidence-base Welbot’s hypotheses (as stated below); (2) to validate, create, and analyse intervention content; and (3) to conduct a pilot test of Welbot.

#### 2.1.1. Stage 1—Hypothesis Validation

This phase involved conducting a literature review in order to evidence-base Welbot’s 3 key hypotheses: (i) sedentary computer-bound workstyles produce unhealthy outcomes (e.g., decline in eyesight, musculoskeletal problems, poor mental wellbeing); (ii) there are strategies and interventions that can produce healthier behaviours in the workplace (e.g., stretching, mindfulness, screen breaks); and (iii) real-time computer prompts can deliver effective interventions that change behaviours. Stage 1 of the project culminated in a comprehensive report of the evidence, which helped to guide refinements and future developments with the digital intervention.

#### 2.1.2. Stage 2—Content Validation, Creation, and Analysis

This section formed the largest part of the project and its purpose was threefold: to validate, create, and analyse Welbot content. Firstly, the research team validated previously developed content including stretches, exercises, and mindfulness nudges. This involved providing a quality score, correlating specific nudges with the evidence, and suggesting recommendations for improvement. Secondly, the research team created new content for the digital intervention, which resulted in approximately 532 new nudges. The final phase of Stage 2 was content analysis. This involved collating all content into 4-week progressive journeys that were either more physically orientated (e.g., “Stand Up, Sit Less, and Move More” or “Less Time on Screens”) or mentally orientated (e.g., “Reduce Stress”, or “Reduce Procrastination”).

#### 2.1.3. Stage 3—Pilot Testing

The final stage, which is the focus of this paper, was a pilot test of Welbot. This involved assessing the efficacy of Welbot on improving sedentary behaviour, physical, and mental wellbeing and exploring participants’ experiences of using Welbot, with quantitative and qualitative findings presented.

#### 2.1.4. Welbot within the Current Study

Although Welbot is a personalised digital intervention, within the current study, a standardised 2-week progressive journey was used to ensure that each user had the same experience of the digital intervention. Nevertheless, it was personalised to this specific group of participants by tailoring content in line with their working hours and work environment (i.e., office-based, sitting desks, shared offices, availability of green spaces, number of stairs/elevators etc.). The nudges within this 2-week journey were randomised by week and were comprised of the following: 60% seated/standing exercises and stretches nudges, 14% relaxed breathing nudges, 11% screen breaks and/or improved work environment nudges, 8% mindfulness nudges, and 7% hydration nudges.

### 2.2. Design

The design of the pilot testing was a single arm repeated measures study over a three-week period (1-week baseline, 2-week intervention) using Welbot. The 2-week intervention period was chosen as it was a pilot study designed to provide quick qualitative and quantitative feedback on the efficacy of the Welbot intervention including areas of improvement to focus on within the next version. The sample size was chosen based on the recommendation from [20] for pilot trial sample sizes with an expected small effect. The primary dependent variable was sedentary behavior, with health and mental wellbeing as a secondary variable. The variables comprising mental wellbeing were: depression, anxiety, stress, procrastination, wellbeing, and work engagement.

### 2.3. Procedure

Ethical approval (04/22/10/18/A) was obtained from the University Ethics Committee. Participants were recruited through opportunity sampling via posters distributed around the University campus, emails sent directly to staff members, and University bulletin board advertisements. Interested participants were eligible if they met the following criteria: (i) aged between 22–65-years; (ii) full time or part time employee of the University; (iii) worked in an office-based environment; (iv) understood the requirements of the study; and (v) had no physical health issue (e.g., severe back pain) that would affect their ability to alter their sedentary behaviour.

Participants, firstly, provided informed consent and then completed an online Qualtrics questionnaire (see outcome measures). Following completion of the pre-intervention online questionnaire, participants were then provided access to the digital intervention and instructed to use this product for 2 weeks. Objective measurement of sedentary behaviour was obtained on a subset of participants (*n* = 6) for a continuous 3-week period (1-week baseline, 2-week intervention). Following the 2-week intervention period, participants completed the online post-intervention questionnaire. An additional subset of participants (*n* = 6) were then asked to take part in an audio recorded one-to-one semi-structured interview to explore their experiences of using the Welbot digital intervention.

### 2.4. Outcome Measures

#### 2.4.1. Demographics

A series of demographics were collected, including: gender, age, height, weight, ethnicity, health status, working status, and occupation.

#### 2.4.2. Primary Outcome Measures

The Occupational Sitting and Physical Activity Questionnaire (OSPAQ) was used to subjectively measure behaviour [21]. This questionnaire measured the time spent sitting, standing, walking, and doing physically demanding tasks or heavy labour. The OSPAQ shows a moderate criterion validity for sitting (ρ = 0.65) and excellent test-retest reliability (ICC = 0.89) [21]. Comparison of sitting measures with accelerometers showed higher Spearman correlations for the OSPAQ (*r* = 0.65) compared to a modified version of the MONICA Optional Study on Physical Activity Questionnaire (modified MOSPA-Q). Criterion validity correlations for occupational standing and walking measures were comparable for both instruments with accelerometers (standing: *r* = 0.49; walking: *r* = 0.27–0.29).

Sedentary behaviour was measured objectively using the ActivPAL accelerometer [22]. Participants were asked to wear the ActivPAL for 24 h per day for 3 weeks continuously. The ActivPAL was made waterproof and attached to the midline anterior aspect of the upper thigh using tegaderm tape. In addition, participants were asked to complete a diary recording the times they started and finished work each day as well as their bed and wake times. Participants were included in the waking day analysis if they provided a complete wear time diary and at least 3 days of valid wear per week (i.e., 3 days with more than 600 min of wear per day) [23]. ActivPAL event files were created using the ActivPAL software provided by the manufacturer. A specialised macro (available from XJ upon request) was then used to calculate time spent sitting, standing, or stepping per waking and working day as well as the average % of time (to account for differences in wake/work times). Moreover, to measure any changes in patterns of sedentary behaviour, the number of bouts between 10–19.99 min, 20–29.99 min, and >30 min were calculated.

#### 2.4.3. Secondary Outcome Measures

The Warwick-Edinburgh Wellbeing Scale (WEMWBS) was used to subjectively measure mental wellbeing [24]. This scale has 14-items and is scored on 5-point scales (*1 = None of the time–5 = All of the time*), where higher scores indicate higher levels of mental wellbeing. WEMWBS has demonstrated both high internal consistency (Cronbach’s α = 0.89–0.91) and high test-retest reliability (0.83) [24].

Five items from the 20-item General Procrastination Scale (GPS) were used to measure participant’s levels of procrastination (i.e., the tendency to delay starting an important task for a more trivial or less important task) [25]. This scale was scored on 5-point scales (*1 = Does not describe me at all–5 = Describes me a great deal)*, such that higher scores reflect higher levels of procrastination. The full version of this scale has shown good levels of reliability (Cronbach’s α = 0.87) in previous research [26].

The Depression, Anxiety, and Stress Scale (DASS-21) was used as a measure of mental health [27]. It is a 21-item measure that is scored via 4-point scales (*0 = Did not apply at all–3 = Applied to me very much, or most of the time)*, whereby higher scores reflect higher levels of distress. This measure calculates scores for depression, anxiety, and stress. Previous research has shown the DASS-21 to have good psychometric properties (Cronbach’s α = 0.82–0.88) [28].

The Utrecht Work Engagement Scale (UWES) was utilised to assess the extent to which participants reported feeling positive, fulfilled, and in a work-related state of mind, characterised by vigour, dedication, and absorption [29]. It is a 17-item measure scored via 7-point scales (*0 = Never–6 = Always*), whereby higher scores exhibit higher levels of work engagement. The UWES calculates a total score plus 3 subscale scores, including: (i) vigour (i.e., high levels of energy, persistence, and resilience towards one’s work activities), dedication (i.e., high levels of enthusiasm and investment in one’s work, with a sense that it has meaning and purpose), and absorption (i.e., being fully and happily engrossed in one’s work). This scale has documented high psychometric properties (Cronbach’s α = 0.80–0.90) in research [29].

### 2.5. Analysis

Descriptive analyses were conducted on questionnaire and ActivPAL data. Paired *t*-tests were used to examine changes between baseline and follow-up for all of the questionnaire outcomes. A Friedman non-parametric test was conducted to examine changes between baseline, follow-up week 1, and follow-up week 2 in time spent sedentary, standing, and stepping and the number of bouts of sedentary behaviour as measured by the ActivPAL. All statistical analyses were conducted in SPSS version 25 [30] and the level of statistical significance was set at *p* < 0.05.

Interviews were examined independently by one member of the research team (AMG) and subsequently cross-checked by an additional member of the research team (AK). Interviews were analysed with a view to gaining a contextualised understanding of participants’ experiences of using the Welbot intervention using a thematic analysis framework [31]. Meaning units that included words, sentences, or phrases relating to the research aim were identified within each transcript and were grouped together based on similar meanings, creating first-order themes [32]. Relationships between these first-order themes were then identified, resulting in overall themes. To ensure quality in the analysis, the research team discussed the thematic analysis and agreed on the themes developed and quotations from the original transcripts are used in the presentation of the findings to demonstrate that the first-order themes did emerge from the interview data.

## 3. Results

### 3.1. Participant Characteristics

Forty-one staff members from a University in the UK participated in this study. A subset of participants (*n* = 6) wore an ActivPAL as part of the study in order to analyse activity data; while another subset of participants (*n* = 6) completed a qualitative semi-structured interview to ascertain user’s experiences of utilising the Welbot intervention. See Table 1 for participant characteristics. No significant differences were found in participant characteristics between the whole sample, ActivPAL, and interview participants. It was noted though that while non-significant, less administrative and secretarial staff and more associate professional and technical staff took part in the interviews.

### 3.2. Self-Reported Sitting Time and Physical Activity

Thirty-nine participants completed the occupational sitting and physical activity questionnaire. Results are presented in Table 2.

### 3.3. Whole Day Sedentary Behaviour

Six participants provided valid ActivPAL data for three weeks. Results are displayed in Table 3.

### 3.4. Working Day Sedentary Behaviour

Intervention results during the working day are displayed in Table 4.

### 3.5. Health and Wellbeing Outcomes

Forty-one participants completed the health and wellbeing questionnaires and results are shown in Table 5. Briefly, the changes for all outcomes were in the expected direction. However, only DASS scores showed significant changes (*depression:* mild–normal; *anxiety:* moderate–mild; *stress:* mild–mild).

### 3.6. Individual Interviews

Thematic analysis of the interviews identified 10 first-order themes and three overall themes relating to participants’ experiences of using the Welbot intervention. Participant quotations highlighting each first-order theme and the three overall themes relating to positive attributes of the Welbot intervention, negative attributes of the Welbot intervention, and suggested improvements are shown in Table 6.

## 4. Discussion

This study evaluated the effectiveness of a new and innovative digital intervention—“Welbot”—that aims to improve physical and mental wellbeing at work. Following the intervention, participants self-reported significantly less time sitting and more time standing and objectively recorded more steps at the week-1 follow-up (compared to baseline) and less prolonged sedentary behaviour (>30 min). A number of outcomes relating to objective measurement of sedentary behaviour (sedentary time, standing time, bouts of sedentary behaviour) showed a trend towards improvement at the week-1 follow-up, however these improvements were not maintained by the week-2 follow-up and hence were not coherent with subjective measures. These trends for improvements in sedentary outcomes are comparable with previous research [33] reporting that computer-delivered prompt interventions are an effective way to not only reduce sedentary behaviour, but also promote physical activity in desk-bound employees. The finding that improvements in sedentary outcomes were not maintained by week 2 follow-up is also consistent with a recent systematic review of mobile health interventions to promote physical activity and reduce sedentary behaviour in the workplace, which emphasised the need to explore the reasons for decline in engagement with interventions [15]. Future research would benefit from including a larger sample of participants, particularly for the objective ActivPAL component. Future research with the Welbot intervention should explore how participants use the programme over time and which components are most effective in promoting sedentary behaviour change to enhance the effect of the intervention.

Following the intervention, participants reported significant improvements in depression, anxiety, and stress; post-intervention all participants were in the “normal” to “mild” range. Such results are congruent with previous research, which have exhibited digital interventions as an effective avenue for improving employees’ mental wellbeing at work [16], particularly for employees’ who are experiencing less complex psychological difficulties. This is crucial given the significant levels of psychological difficulties in the current working population, with occupational-related stress, anxiety, and/or depression affecting 526,000 employees living in Britain in 2016/17 [34]. Perhaps Welbot functions to improve employees’ depression, anxiety, and stress via its emphasis on mindfulness. An abundance of previous literature has highlighted the benefits of mindfulness in the workplace, with improvements in depression, fatigue, stress, anxiety, burnout, job performance, and work–life equilibrium [35,36,37,38]. Contrastingly, such improvements may also be resultant from its focus on decreasing sedentary behaviour. Indeed, research has exhibited that prolonged occupational sitting is associated with higher levels of psychological distress [39]. Hence, a nudge-based digital intervention that aims to reduce this adverse behaviour may secondarily also improve mental wellbeing. Alternatively, these improvements could merely be due to the simple act of taking a break from work tasks. Such breaks have been shown to reduce levels of fatigue and improve employees’ sense of vitality [40]. Regardless of what specific elements are driving this change, it appears that Welbot, as a digital intervention, can enhance some areas of psychological functioning (i.e., depression, anxiety, and stress). Contrastingly, results demonstrated that levels of mental wellbeing failed to significantly change following the intervention. Hence, although it is positive that the digital intervention did not impair functioning, this limits our ability to make any firm conclusions about Welbot’s overall impact on mental wellbeing outcomes. Therefore, further research utilising larger sample sizes and a control group is needed.

Results exhibited that levels of procrastination and work engagement failed to significantly alter post-intervention. This is a novel finding in the emerging field of nudge-based digital interventions and although it was somewhat unexpected, it indicates that the introduction of Welbot to the workplace did not impair occupational functioning. In particular, the lack of change with regards to employees’ work engagement is of importance. Despite Welbot interrupting employees every 40-min and asking them to participate in a nudge, this “time-out” from work tasks did not correspond to a detriment in work engagement. Previous research has shown that digital interventions are in fact capable of enhancing job performance and employee health [16,41]. Therefore, although it is positive that Welbot did not impair employees’ functioning in these areas, prospective research should now focus on investigating how the digital intervention can be further refined to optimise its effect on occupational functioning.

Several positive attributes of the digital intervention were identified by the participants, including a perception of positive behaviour change as a result of using Welbot. These perceptions were partially supported by the quantitative findings where self-reported sitting time during the working day significantly decreased over the two weeks. Participants also reported that their awareness of unhealthy behaviours at work increased as a result of using the digital intervention, which has been echoed in a recent systematic review exploring factors affecting patient and public engagement with digital interventions [42]. In relation to the negative attributes of the digital intervention, feeling self-conscious whilst carrying out the activities at their desk was reported in previous studies that specifically used health interventions in the workplace. Indeed, research has found that not having a private space within the workplace to access a digital mental health intervention and feeling exposed using the intervention whilst sat at their desk were barriers to engagement [43]. Whilst participants reported on the positive and negative attributes of Welbot, they did offer useful suggestions for improvements to the digital health intervention. Several participants advocated for Welbot to allow individualised preferences in relation to the frequency and timing of nudges and the option to sync these to online calendars. Personalised tailoring of information within digital health interventions has been reported by users in previous research as an important facilitator of continued engagement [42,43,44]. Indeed, more recent versions of Welbot than what was used in the current study offer this function. Nevertheless, the qualitative component of this pilot study utilised a small sample size (*n* = 6), which may not necessarily be representative of the sample as a whole. Therefore, further qualitative research into people’s experiences of using Welbot are needed in order to make it as practical, accessible, and user-friendly as possible.

The main limitation of this pilot study was that it utilised a small sample size (*n* = 41), which was even further compromised in the ActivPAL (*n* = 6) and qualitative (*n* = 6) components. This largely limits the interpretation of findings and hence, caution is needed. Therefore, future larger studies would benefit from conducting a priori power analysis to ensure an adequate number of participants to detect an effect and to explore the potential efficacy of Welbot in reducing sedentary behaviour and improving wellbeing in the workplace. Other limitations to this study relate to the participant demographic characteristics and study design. The majority of participants were white, female, full time workers. Therefore, prospective studies utilising larger sample sizes (particularly for the objective ActivPAL component) and more diverse participant demographical characteristics are needed. The study used a single-arm repeated measure design and there was no control group. This limits our ability to attribute any improvements in wellbeing to the digital intervention as it is plausible that simply delivering regular notifications to users without any associated task may perform just as well. Therefore, further research adopting a two arm, repeated measures design with a control group and participants randomly allocated to each arm of the study is essential. A further limitation is the short intervention period (i.e., 2 weeks), hence prospective research would benefit from using longer intervention periods. It would also be helpful to have a longer follow up period (e.g., 6 months) to ascertain whether any benefits of using the digital intervention are maintained over time.

A further limitation may be the use of the ActivPAL research device, which may have encouraged participants to change their behaviour. The possibility of aligning Welbot with personal activity trackers could be explored as a means of capturing and integrating such feedback. Finally, it remains unclear which elements of this multi-component intervention are most important for maximising health and wellbeing gains. An in-depth analysis of user engagement patterns alongside the use of standardised outcome measures and improved reporting of “active” components of Welbot would enhance the future evaluation of this digital intervention.

## 5. Conclusions

Overall, findings from this pilot study provide indications of the potential positive impact Welbot may have on employees’ physical and mental wellbeing. Adopting a multi-method approach, using objective activity measurement and standardised outcome measures alongside qualitative data detailing users’ experiences of Welbot allowed more holistic insight into the promise of this digital intervention. Nevertheless, as this study was only a pilot study with a small sample size and no control group, caution is needed when interpreting findings and further research is required. The findings also helped to outline some possible refinements to the digital intervention which may help to improve user experiences. Capturing data as to how participants use and engage with Welbot is necessary. Detailing the amount of time users spend on each component of the application and which components were used the most is important and could be explored in future work. Overall, these results provide an initial snapshot of the potential effectiveness of Welbot and a useful baseline for further intervention development and research.

## Figures and Tables

**Figure 1 ijerph-17-05763-f001:**
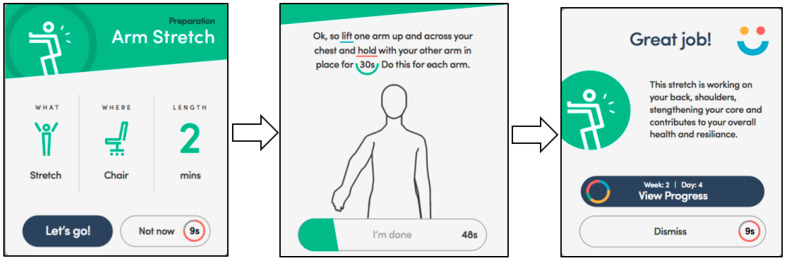
Example of nudge delivered in the intervention (Note: Permission to include this Figure was granted by Welbot [19]).

**Table 1 ijerph-17-05763-t001:** Participant Characteristics.

	Whole Sample (*n* = 41)	ActivPAL Sample (*n* = 6)	Interview Sample (*n* = 6)
*Demographics*			
*M* age (SD)	43 years (10.40)	46 years (8.26)	42 years (13.10)
Age range	22–63 years	30–53 years	29–58 years
*M* BMI (SD)	24.82 (3.79)	27.41 (8.68)	23.07 (2.46)
Males (%)	6 (14.63%)	1 (16.67%)	1 (16.67%)
Females (%)	35 (85.37%)	5 (83.33%)	5 (83.33%)
*Ethnicity*			
White	95.12%	100%	100%
Other	4.88%	0%	0%
*Working hours*			
Part-time (%)	2 (4.88%)	0%	0%
Full-time (%)	39 (95.12%)	100%	100%
*Occupations*			
Managers, directors, and senior officials	14.63%	0%	0%
Professional occupations	39.02%	50.00%	50.00%
Associate professional and technical occupations	9.76%	16.67%	33.33%
Administrative and secretarial occupations	36.59%	33.33%	16.67%

**Table 2 ijerph-17-05763-t002:** Occupational sitting and Physical Activity Questionnaire (mean ± SD; *n* = 39).

	Baseline	Follow-Up	*p*-Value ^1^
Sitting time (h/working day)	6.5 (1.2)	5.6 (1.5)	<001
Standing time (h/working day)	0.5 (0.4)	0.8 (0.8)	0.009
Stepping time (h/working day)	0.7 (0.4)	0.9 (0.6)	0.175
Heavy labour (h/working day)	0.03 (0.1)	0.04 (0.2)	0.696

^1^ Paired sample *t*-test for baseline and follow-up measures.

**Table 3 ijerph-17-05763-t003:** Intervention outcomes for the whole day (mean ± SD; *n* = 6).

	Baseline	Follow Up Week 1	Follow Up Week 2	*p*-Value ^1^
Sedentary time (%)	59.8 (6.4)	57.2 (1.4)	59.7 (8.7)	0.846
Standing time (%)	25.2 (5.0)	26.2 (6.7)	24.8 (6.1)	0.846
Stepping time (%)	14.9 (2.7)	16.6 (3.3)	15.5 (3.5)	0.069
Bouts 10–19.99 min	6.2 (2.3)	6.0 (2.1)	5.7 (2.9)	0.607
Bouts 20–20.99 min	3.3 (0.8)	2.7 (0.6)	2.7 (0.7)	0.200
Bouts > 30 min	5.0 (1.2)	4.8 (1.9)	5.3 (2.1)	0.827

^1^ Friedman’s non-parametric test for between time point differences.

**Table 4 ijerph-17-05763-t004:** Intervention outcomes for the working day (mean ± SD; *n* = 6).

	Baseline	Follow Up Week 1	Follow Up Week 2	*p*-Value ^1^
Sedentary time (%)	60.6 (8.4)	54.8 (10.8)	60.5 (7.4)	0.846
Standing time (%)	23.6 (5.9)	25.8 (8.0)	23.9 (5.6)	0.513
Stepping time (%)	15.8 (4.4)	19.4 (4.2) ^a^	15.6 (2.9)	0.042
Bouts 10–19.99 min	3.1 (1.1)	3.0 (1.5)	2.5 (1.5)	0.607
Bouts 20–20.99 min	1.9 (0.7)	1.2 (0.5)	1.4 (0.7)	0.311
Bouts > 30 min	2.8 (1.2)	2.5 (1.3)	2.8 (1.4)	0.467

^1^ Friedman’s non-parametric test for between time point differences; ^a^ significantly different from baseline and follow up week 2.

**Table 5 ijerph-17-05763-t005:** Health and wellbeing questionnaire (mean ± SD; *n* = 41).

	Baseline	Follow-Up	*p*-Value ^1^
WEMWBS	49.05 (7.44)	50.07 (6.49)	0.265
GPS-S score	14.39 (4.78)	13.41 (4.60)	0.060
UWES total score	3.53 (0.69)	3.63 (0.86)	0.237
UWES vigour score	3.53 (0.66)	3.63 (0.88)	0.238
UWES dedication score	3.72 (0.93)	3.75 (1.03)	0.751
UWES absorption score	3.39 (0.72)	3.51 (0.85)	0.111
DASS depression score	11.35 (5.44)	9.25 (2.28)	0.051
DASS anxiety score	10.25 (3.81)	8.55 (2.01)	0.004
DASS stress score	14.85 (6.52)	12.20 (3.98)	0.008

^1^ Paired sample *t*-test for baseline and follow-up measures.

**Table 6 ijerph-17-05763-t006:** Overall themes, first-order themes, and participant quotations (*n* = 6).

Overarching Theme	Subtheme	Example
*Positive Attributes*	Achieved behavioural change (*n* = 6)	*“I liked the prompts to get up and walk and stretch, which is stuff I wouldn’t have done necessarily otherwise”*
	Appropriate timing/duration of nudges (*n* = 3)	*“I would say the nudges were enough, I wouldn’t have them anymore or have them spaced out longer”*
	Enhanced awareness of unhealthy behaviours (*n* = 4)	*“It made me realise that I don’t drink enough water”*
	Continued use of app post-intervention (*n* = 5)	*“I still have it installed on my computer and this morning I did two of the stretches”*
*Negative Attributes*	Inappropriate timing/duration of nudges (*n* = 3)	*“I found it asked me an awful lot about how many drinks of water I have had... it would ask me at 10 am and I’ve only been awake a few hours so I haven’t had all my glasses yet!”*
	Technical issues (*n* = 5)	*“I had the same things popping up again and again”*
	Mindfulness nudges too time-consuming (*n* = 5)	*“The mindfulness ones I didn’t like so much... they kind of got in the way a little bit of me doing the rest of my work”*
	Self-conscious doing exercises at desk (*n* = 2)	*“I was thinking everyone is going to be wondering what I’m doing standing up and doing these stretches. I felt a bit self-conscious”*
*Improvements*	Sync the app with outlook calendar (*n* = 1)	*“If there was the capacity to have it integrated with your outlook calendar I think that would be a massive improvement”*
	Individualised preferences in the app (*n* = 3)	*“It would have been cool if I was able to say I want lots of break exercises and full body stretches”*

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
