# Peer review of "Pilot Testing of a Nudge-Based Digital Intervention (Welbot) to Improve Sedentary Behaviour and Wellbeing in the Workplace"

_ijerph, 2020, doi:10.3390/ijerph17165763_

Round 1

Reviewer 1 Report

Overall

This is an interesting paper reporting the results of a small, non-controlled workplace intervention to reduce sitting time. Authors are commended for including both quantitative and qualitative methods and results in the paper. However, details about the intervention are lacking and power is not reported making it difficult to put the results into context with the larger literature set. 

Introduction

Paragraph 1 and 4 address interventions in the workplace.  Paragraphs 2 and 3 address sedentary behavior.  Consider placing paragraphs 1 and 4 together, it would provide more cohesive flow to the introduction.

It is not clear why another intervention needs to be developed, there are digital interventions available, why is yours needed?

You measure a number of secondary factors, but there is no/limited rationale in the introduction for these measures.

Methods

It is not clear how this intervention is personalized to each user.

Lines 86-87.  How often do the nudges occur (what is the range… anywhere from every 15 minutes to every 4 hours (the discussion suggests every 40 minutes… this would mean they receive 12 nudges/workday)?  How is that decided?  Does the participant decide this? Who decides what nudges each person receives, or is it standardized?  If some of the nudges, such as the example in the paper, do not ask the person to move their lower body, you would not expect the participants to decrease their sedentary behavior.  More information on the intervention is needed.

Line 155-166: It is not clear whether the ActivPal was worn 24 hours/day or only during waking hours.  Please clarify.  Also, provide a reference for valid wear time.

Two weeks is a very short time for an intervention.  The authors should provide rationale for such a short time period and supporting evidence that the outcomes will change during this time.

Please describe the subject inclusion/exclusion criteria and the sample size calculations.

Results

Were there differences in the groups (Table 1- whole groups, Activpal, interviews)?  This is important to help the reader know whether the small groups are representative of the whole.

If the participants were able to choose function on the intervention, please report these.  For instance, if they were able to choose how often they received nudges and what those nudges were, please report those details.

The text in section 3.2 and table 2 are repetitive. Consider presenting the data in the text or table in a different manner (e.g., percent of workday), or remove the table or text.

Line 237.  There is a semi-colon and period at the end of the table footnote.

Please provide the power for the analyses.

Some of the text in Figure 2 is cut out/off.

Discussion/Conclusion

Line 271. It would be clearer if the authors were explicit that you are comparing week 1 follow-up to baseline (I am assuming).

Lines 295-296.  This is a strong statement given that the WEMWBS did not significantly change. Please revise to better represent the results of your study.

There is minimal discussion of how these results compare with other workplace interventions to reduce sitting.  Were the results similar?  What was different in this study?

Author Response

Please see attached response.

Reviewer 2 Report

This pilot study aims to provide operators with a tool for promoting health in the workplace. From this point of life the effort of the authors is certainly appreciable.

The sample examined is very small and this represents the main limitation of the study. The statistical difference on 6 cases can be seen with difficulty, confounding factors cannot be considered. This severely limits discussion of results and conclusions.

We agree that a power analysis is needed to determine the appropriate sample size. If the authors had made this calculation, they would not have presented so few cases.

The main advice could be to continue the research and present the analysis  of a larger number of people. Since this is not possible in the time frame for reviewing this article, authors can at least report this issue.

Reviewer 3 Report

In this well-written article, Haile et al. pilot tested the use of a digital intervention (Welbot app) to reduce sedentary behavior and improve physical and mental wellbeing in the workplace. The study included 41 university staff that was largely comprised of white, non-obese, female, and full time workers. Study duration was 3 weeks (1 week baseline, 2 weeks intervention). A number of various scoring methods were used with interview feedback from a small portion of participants. In the end, the authors claim that the findings in this work support the use of Welbot to improve physical and mental wellbeing in the workplace.

Major comments:

  • The authors claim in the title and throughout the manuscript that Welbot is a “personalized” digital intervention. It is certainly a digital intervention, although as the feedback from participants and the authors share in the discussion, this is in fact not a personalized or individualized approach. While a digital intervention indeed, suggesting this is a personalized/individualized approach is an over-reaching claim and should not be made unless further justified. In fact, it is unclear is a
  • The authors conclude that the “findings support the use of Welbot as an acceptable and practical way to improve employee’ physical and mental wellbeing at work.” Unfortunately, this is not true, and the authors admit as much in the discussion. The same size is too small and lacks diversity. Moreover, the data collected demonstrates that any gains in the first week were not maintained, or returned to baseline, after week 2. It may perhaps be the case that if the study was performed longer than 2 weeks, with a control arm, the results may have been in favor. In addition, it is also unclear if a simple notification/alert throughout the day (i.e., at similar time intervals as the ones used in this study) without any associated task would perform just as well. Thus, while support in favor of intervention implementation at this point is not lacking, support in favor of a larger, more diverse prospective study with longer follow-up and additional arms is supported. Again, the authors should scale back on their claims.

Minor comments:

  • Is it possible to know how many “nudges” were actually completed and ignored? This would be helpful. You may find that those that did not ignore it continued to do well after 1 week.
  • Figure 2 should be reviewed. Some words are cut off in the text boxes.

Despite the aforementioned limitations, the authors provided a detailed, well-written, and thoughtful analysis of a unique intervention to improve wellbeing and should be commended – well done!

Reviewer 4 Report

This paper describes a pilot study to check for the efficacy of the digital intervention Welbot.

The authors are very positive about the Welbot intervention and they conclude  'this digital intervention to have promise as a method for improving various aspects of employee health'

I am not convinced.

There were differences on the Depression, Anxiety and Stress Scale. The authors do not reflect on the relevance of this difference. What is the interpretation of the scores? Is there a relevant well being benefit? If I am correct the scores are low, no signs of depression and anxiety at the start. Is it relevant to see a reduction at these levels?

The results for subjective and objective sitting time / sedentary time are not consistent. I should be worried when the objective measure does not show a reduction in sedentary time. Especially since, sedentary behaviour is an independent risk factor for obesity and chronic disease. The welbot intervention was not able to reduce this risk.

There were no differences on The Warwick-Edinburgh Wellbeing Scale, General Procrastination Scale and The Utrecht Work Engagement Scale.

My conclusion with the same data would be that the intervention is not (yet) good enough. With the recommendation to make a few more iterations in the intervention development.

The authors themselves suggest to improve the intervention but also mention a RCT.

A RCT can wait, make the intervention better and repeat the efficacy study with a control group, extent the duration of the intervention and measure more subjects with the activPAL.

Other points:

  • what is the interpretation of the results for each outcome measure. F.I. What is the interpretation of a WEMWBS of 49.05 ? Is this a risk where you want to intervene on
  • What make the intervention personalized? I could not find anything in the paper on this topic except that it would be nice to have it personalized in the future......
  • At baseline, participants accumulated 5.0 ± 1.2 bouts of sedentary behaviour greater than 30 minutes per day, this decreased significantly to 4.8 ± 1.9 bouts per day during week 1 of the intervention, and increased to 5.3 ± 2.1 bouts par day during week 2 of the intervention. Is significantly correct?

Round 2

Reviewer 2 Report

The authors followed the indications, and improved the manuscript

Author Response

Dear Reviewer,

Thank you for your thoughtful comments and recommended changes. Please see the attached cover letter detailing all changes that have been made.

kind regards,

Nicola

Reviewer 4 Report

 Compliments.

The authors have adequately addressed my comments.

Author Response

Dear Reviewer,

Thank you for your thoughtful comments and suggested amendments. Please see the attached cover letter.

Kind regards,

Nicola
